# Emission Characteristics and Potential Exposure Assessment of Aerosols and Ultrafine Particles at Two French Airports

Sébastien Artous [1,*], Eric Zimmermann [1], Cécile Philippot [1], Sébastien Jacquinot [1], Dominique Locatelli [1], Adeline Tarantini [2], Carey Suehs [3], Léa Touri [4] and Simon Clavaguera [1]

1   University Grenoble Alpes, CEA, Liten, DTNM, 38000 Grenoble, France; eric.zimmermann@cea.fr (E.Z.); cecile.philippot@cea.fr (C.P.); sebastien.jacquinot@cea.fr (S.J.); simon.clavaguera@cea.fr (S.C.)
2   University Grenoble Alpes, CEA, Nanosafety Plateform (PNS), Laboratory of Medical Biology (LBM), 38000 Grenoble, France; adeline.tarantini@cea.fr
3   Departments of Medical Information and Respiratory Diseases, University Montpellier, CHU Montpellier, 34295 Montpellier, France; careysuehs@protonmail.com
4   Air France Aéronautic, Occupational Health Department, 95747 Roissy Charles de Gaulle CEDEX, France; sabytouri@hotmail.com
*   Correspondence: sebastien.artous@cea.fr

**Abstract:** Airports are significant contributors of atmospheric pollutant aerosols, namely ultrafine particles (UFPs). This study characterizes the particle number concentration (PNC), the median particle size ($d_{mn50}$), and the metallic composition of medium-haul area and engine aerosols at two French airports (Paris-CDG and Marseille). This study followed the standard operating procedures for characterizing aerosol emissions from 5 nm to 8 μm (OECD, 2015; EN 17058:2018). It allows determining which are the specific parameters directly related to the emission sources and their contribution to the overall aerosols measured at workplace in airports. The particulate emissions observed during aircraft engine start-up were ~19× higher than the average airborne concentration. The particle size distributions remained mostly <250 nm with $d_{mn50}$ < 100 nm (showing a specificity for the medium-haul area with an average $d_{mn50}$ of ~12 nm). The $d_{mn50}$ can be used to distinguish emission peaks due to aircrafts ($d_{mn50}$~15 nm) from those due to apron vehicle activities ($d_{mn50}$ > 20 nm). Chemical elements (titanium and zinc) were identified as potential tracers of aircraft emissions and occurred mainly at the micrometric scale. For aircraft engine emissions, UFPs are mainly due to fuel combustion with the presence of carbon/oxygen. The study concludes with suggestions for future research to extend on the findings presented.

**Keywords:** aerosols; ultrafine particles; aircraft emission; elemental carbon; zinc; titanium





## 1. Introduction

Airports represent a significant source of air pollutants, especially aerosols [1–4] and ultrafine particles (UFPs) [5]. Combustion within motor/jet engines results in UFPs containing emissions demonstrably associated with diverse health repercussions, including respiratory and cardiopulmonary effects as well as lung cancer [6]. In 2012, the International Agency for Research on Cancer (IARC, at the World Health Organization) classified diesel engine exhausts as carcinogenic to humans [7]. The particle profiles resulting from aircraft kerosene combustion and diesel motors overlap [8]. Previous studies have shown that airports are a significant source of such emissions, with human exposure being a significant health concern [8,9]. Schlenker et al. demonstrate the correlation between airport pollution and respiratory health. Specifically, they found that a one standard deviation increase in airport pollution is responsible for approximately one third of daily admissions for asthma problems [10].

UFPs are a subset of aerosols with aerodynamic diameters < 100 nm. In terms of mass, they represent a fraction of airport-emitted aerosols; however, they represent the majority

of emissions when expressed as particle numbers. Desvergne et al. showed that the median PNC is 18 times higher on the apron, with $1.5 \times 10^5$ particles per cubic centimeter (p/cm$^3$), than inside the airport offices, with $8.3 \times 10^3$ p/cm$^3$ [11]. In addition, this fraction can penetrate deeper into the respiratory tract during inhalation [12]. The review performed by Bendtsen et al. [8] highlights that few studies attempt to correlate airport worker lung function with aerosol exposure and especially with UFPs.

Concerning the characterization of emissions, a few previous studies [1–4] show that aircraft emissions are primarily composed of high concentrations of aerosols with a diameter of less than 20 nm. The concentrations on the apron frequently exceed $10^5$ p/cm$^3$ and can occasionally exceed $10^6$ p/cm$^3$; the maximum reached at the Montreal Trudeau Airport is $2 \times 10^6$ p/cm$^3$ [1]. The apron is the location where an aircraft is parked, unboarded or boarded, refueled, and serviced. The main sources in number are reactors [4] and combustion engine vehicles [11]. Desvergne et al. presented an initial hypothesis regarding the influence of the aircraft engines on the mean particle diameter and differences between locations (apron and offices) [11].

The objective of this study, which is part of the French Nanero project [13], is to extend the current knowledge of airport emissions by sampling specific source emissions, airborne aerosols at a medium-haul hub during a high activity period, at the Paris-Charles de Gaulle airport (Paris-CDG) and Marseille, France, to evaluate the personal worker exposure to aerosols. Specific attention was paid to the particle number concentration (PNC) and to the median particle size ($d_{mn50}$ in nanometers) for eight engines and at a medium-haul workstation area at Paris-CDG in October 2018. The aim is also to identify metallic elements within source emissions and airborne aerosols to potentialize the future development of tracer profiles for occupational exposure. The reader should note that a ~6.6 year longitudinal assessment of lung function was also conducted in parallel with this study and is presented in a separate paper [14].

## 2. Materials and Methods

- Study design

This observational study characterized the particle number concentration (PNC) and the median particle size ($d_{mn50}$ in nanometers) of airport aerosols at two levels: (i) specific engine emissions (also known as emission sources and including aircraft and ground vehicles or generators) and (ii) aerosols at a medium-haul workstation area. Aerosol sampling for (i) and (ii) took place in October 2018 at Paris-CDG airport, France, and was performed using trolley-mounted devices that simultaneously analyzed the aerosols entering a single-probe trolley (Figure 1). Device limitations in terms of the particle size ranges they can treat are indicated by box spans over the nanometer log scale (indicated on the left-hand side of the graphic). Devices located on a trolley for simultaneous mobile assessments included a condensation particle counter (CPC), a fast mobility particle sizer spectrometer (FMPS), and an aethelometer (all on the same sampling probe). A compact membrane sampling device (i.e., Sioutas®, SKC, Eighty Four, PA, USA) was useful to characterize the source emissions directly from engines as well as for the sampling at the medium-haul work area. The Sioutas® device collects particles onto four different membranes according to the size.

- Source emissions: engine exhaust sampling

Landings and take-offs interfered with ambient aerosol characterization. It therefore became necessary to focus on representative emission sources to help to elucidate aerosol variation. It follows then that the first part of the study aerosol sampling strategy consisted in characterizing emissions directly from working engines. The latter included a CFM-56-5B engine made by CFM International for a medium-haul aircraft, a GE 90 engine made by GE Aviation for a long-haul aircraft, a less recent Pushback tractor 256, a new-generation Pushback tractor 270, a generator set, a diesel van (Jumper®, Citroën, Saint Ouen sur Seine, France), and a gasoline car (Clio® III, Renault, Boulogne-Billancourt, France). Aircraft

engine emissions (CFM-56-5B and GE 90) were measured in the exhaust gas chamber of the Zephyr Air-France test bench (Figure S1), using 15 m of 7.8 mm internal diameter of an antistatic flexible tube, for sampling from the sampling point to the single-probe trolley. The remaining exhaust sources were sampled in the exhaust cone with the single-probe trolley in the near field (<3 m) of the source emission. The measurement locations were selected to be as representative as possible of apron worker exposure. Both the trolley-mounted and compact device (Sioutas®) presented in Figure 1 were used. Table S1 summarizes the basic meteorological data recorded by the airport weather station during the source emission characterizations.

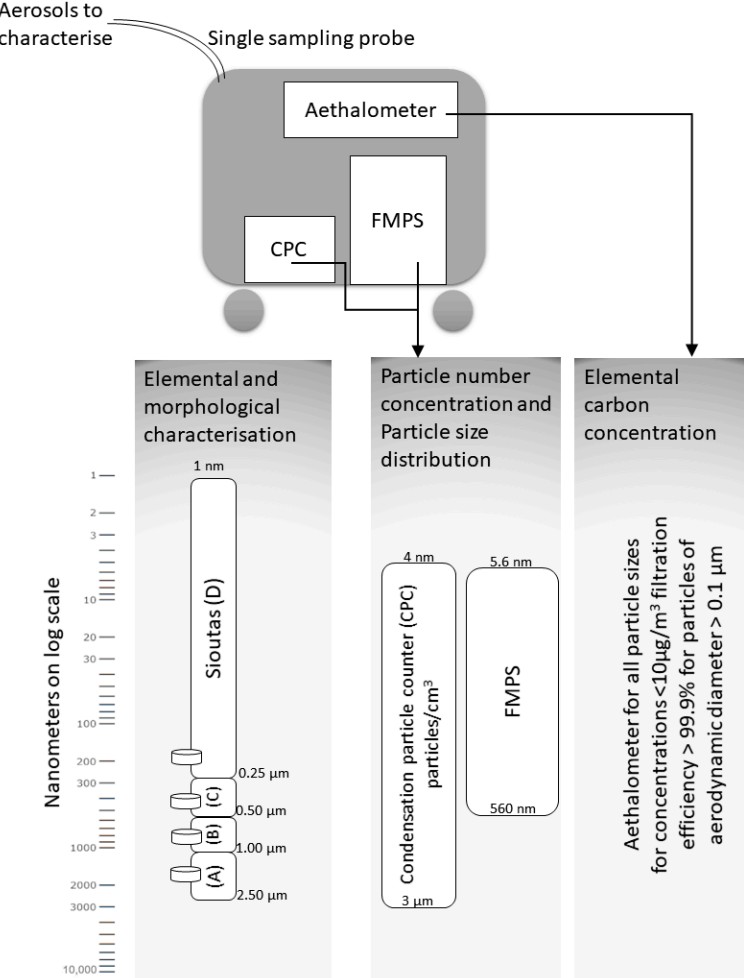

**Figure 1.** An overview of the sampling devices used in the study to determine the particle number concentration (PNC), the median particle size ($d_{mn50}$), and elemental carbon mass concentrations (ECs).

- Aerosol sampling at the medium-haul work area

Continuous aerosol emission measurements in the medium-haul area were performed on 11 October 2018 at parking F30 of terminal 2F at the Paris-CDG airport between 10:00 am and 01:15 pm. This time slot corresponds to a high activity period for the hub (Figure S2). Both the trolley-mounted and compact device presented in Figure 1 were used. Table S1 summarizes the basic meteorological data recorded by the airport weather station during the measurements.

- Single-probe trolley devices for simultaneous aerosol analyses

The devices mounted on the study trolley (Figure 1) simultaneously analyzed the aerosol entering a single sampling probe, as previously described [15]. The particle number concentration (PNC up to $10^7$ p/cm³), for particles ranging from 4 nm to 3 μm in size,

was determined using the butanol-based Model 5416 condensation particle counter (CPC) from GRIMM, DURAG GROUP, Hamburg, Germany. CPC aerosol emission recording was followed by the determination of the median particle size ($d_{mn50}$) and the particle number concentration (PNC up to $2.2 \times 10^7$ p/cm$^3$) via a fast mobility particle size spectrometer (FMPS, Model 3091, TSI, Shoreview, USA); the latter was restricted to particles ranging from 5.6 to 560 nm in size. All real-time recordings were processed using the JMP® 17 software, SAS Institute, Cary, NC, USA. Additionally, an aethalometer (Model AE42-7-ER, Aerosol Magee Scientific, Berkeley, CA, USA) was used to measure elemental carbon (EC) concentrations up to 10 µg/m$^3$.

This study followed the current standard operating procedures for characterizing aerosol emissions at the workplace in the range of 5 nm to 8 µm, the OECD tiered approach [16], and the NF EN 17058:2018 [17]. As the latter is mainly designed for indoor use, the positioning of the ambient measurement was adapted for airport environments by moving the zone of direct influence of the airport apron (5 km from the apron terminal F, Figure S3).

- Compact device for particle collection on membranes

A compact device, Sioutas®, was added to the aerosol analyses for source emissions and at the medium-haul work area. Sioutas® is a personal cascade impactor with four impactor stages plus a final filter that allows the separation and collection of airborne particles for four increasingly smaller size ranges (50% cut-point of each stage at 9 L/min: Stage A: 2.5 µm; Stage B: 1.0 µm; Stage C: 0.50 µm; and Stage D: 0.25 µm). The flow was provided by a Leland Legacy Sample Pump Cat. No. 100-3002 (9 L/min). The particles were collected on Whatman® Nuclepore™ Track-Etched membranes (one per size range), Global Life Sciences Solutions, Malborough, MA, USA, which were later analyzed for the elemental content.

- Membrane analyses

The membranes were analyzed using total reflection X-ray fluorescence (TXRF; performed using a Nanohunter® benchtop spectrometer—Rigaku, Cedar Park, USA) to determine the elements present with an atomic mass equal to or greater than aluminium. The analysis conducted was a semi-quantitative analysis that did not provide the concentration or mass values. The limit of detection (LOD) and limit of quantification (LOQ) for each element were determined as follows: LOD = µ_blank + 3σ_blank and LOQ = µ_blank + 10σ_blank, where µ and σ represent the average and standard deviation, respectively, of the X-ray fluorescence energy quantity of the studied element measured on ten blank filters. This TXRF analysis allowed us to select membranes for the additional scanning electron microscopy coupled with energy dispersive X-ray (SEM/EDX) spectroscopy to further identify the elements with an atomic mass equal to or greater than carbon. The latter was performed using a Model S-5500 SEM (Hitachi High-Tech Corporation, Ibaraki, Japan) combined with a Noran EDX system (Thermo Fisher Scientific®, Waltham, MA, USA), and respected micrometric/nanometric groupings.

## 3. Results

### 3.1. Source Emissions: Engine Exhausts

Figure 2 and Table 1 summarize the real-time measurements (FMPS, PNC, and $d_{mn50}$) and elemental identifications (TXRF and SEM/EDX) during the near-field motor exhaust measurements (i.e., Aircraft engine GE90, Aircraft engine CFM56-5B, Pushback tractor 256, Pushback tractor 270, generator set, gasoline car, and diesel van).

The PNC for the GE90 reactor exhaust (not shown in Figure 2) during take-off exceeded the FMPS quantification limit with a concentration above $2.2 \times 10^7$ p/cm$^3$, which was in the same order of magnitude as that in previously published reports [11,18]. Despite this saturation, the associated median particle size ($d_{mn50}$) can be estimated to be between 15 and 18 nm. The elemental analyses showed that the nanometric fraction is derived mainly from fuel combustion with the presence of primarily carbon/oxygen. Interestingly,

titanium was identified at the micrometric and submicronic scales. The aethalometer, for EC determination, was out of range during the measurement, and this was the case for all of the source emissions measurements. However, this did not affect the study, which focused on the PNC and $d_{mn50}$ emissions from engines for real-time measurements.

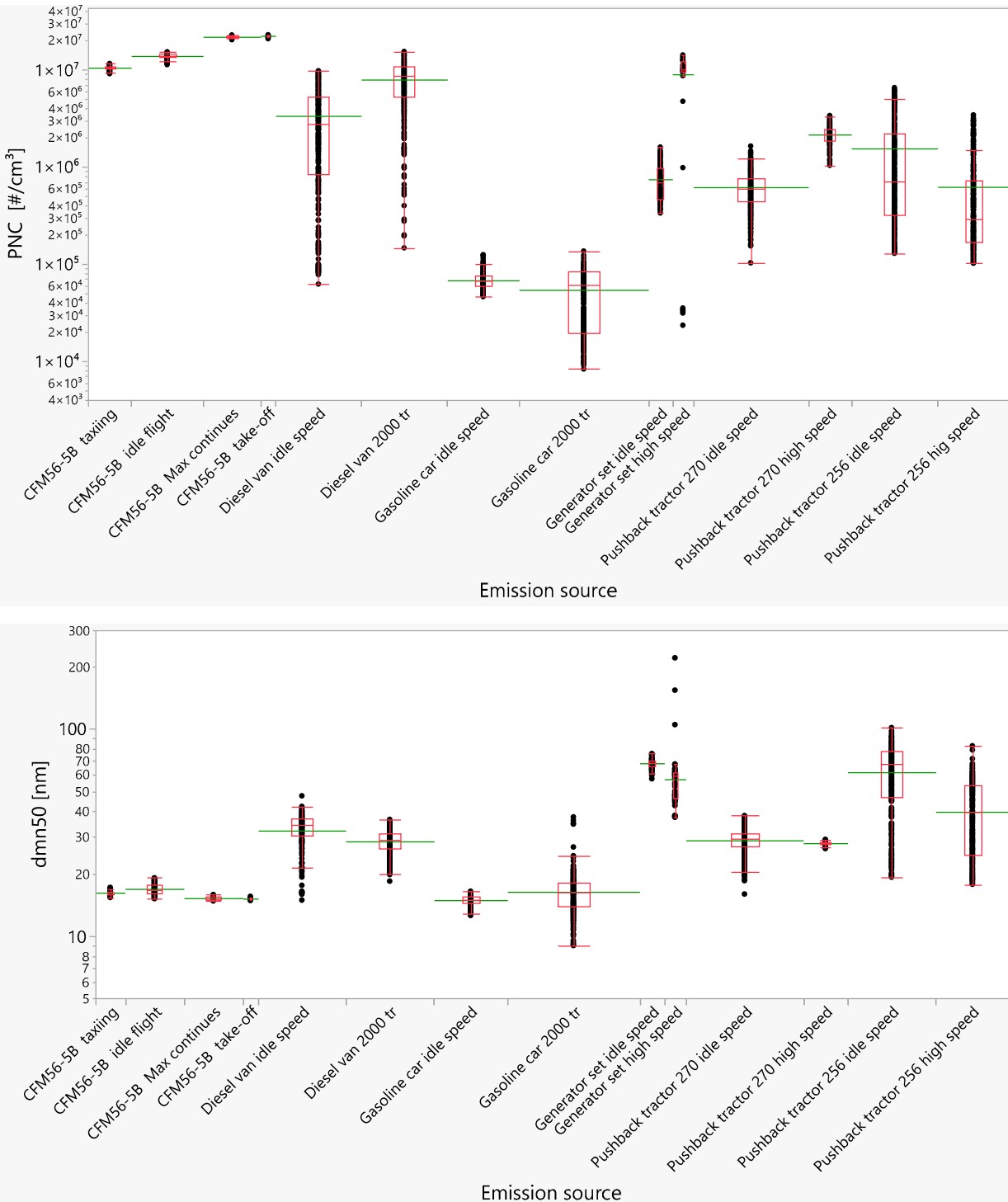

**Figure 2.** Box plots in red and average straight lines in green of the particle number concentration (PNC) and the median particle size ($d_{mn50}$) for the engine exhausts studied (i.e., Aircraft engine CFM56-5B, diesel van, gasoline car, generator set, Pushback tractor 270, and Pushback tractor 256) as a function of the engine rpms.

**Table 1.** Chemical tracer elements for the studied motor exhausts. Elements that were detected below and above the limits of quantification are indicated in standard and bold fonts, respectively, for the total reflection X-ray fluorescence (TXRF).

| | TXRF Chemical Identification | | SEM-EDX Chemical Identification | |
|---|---|---|---|---|
| Sources of Emission | Micrometric Size | Submicronic Size | Micrometric Particles | Nanometric Particles |
| Aircraft engine GE90 (long haul) | Fe, S, Si | Br, Fe, **S**, Ti, | Al, Ca, Cl, O, Si, S, Ti | C, O |
| Aircraft engine CFM56-5B (medium haul) | Br, **Ca**, **Fe**, **K**, Mn, **S**, Si, **Ti** | **Fe**, **S**, Ti, Zn | Al, Ca, Fe, K, Na, O, Si, S, Zn | C, O, Na, S |
| Pushback tractor 256 (old generation) | **Ca**, Fe, K, S, Si, Ti, Zn | Ca, **S**, Fe | Ca, F, O | C, O |
| Pushback tractor 270 (new generation) | **Ca**, **Fe**, S, Si | Br, **Fe**, **S**, Ti, Zn | Cr, O, S | C, O |
| Generator set | -- | -- | -- | -- |
| Gasoline car | Ca, K, **S**, Si | Ca, K, **S**, Si | Cr, Mo, O | C, O |
| Diesel van | | | | |

Al = Aluminum; Br = Bromine; C = Carbon; Ca = Calcium; Cl = Chlorine; Cr = Chromium; F = Fluorine; Fe = Iron; K = Potassium; Mn = Manganese; Mo = Molybdenum; Na = Sodium; O = Oxygen; S = Sulfur; Si = Silicon; Ti = Titanium; TXRF = Total reflection X-ray fluorescence; SEM-EDX = Scanning electron microscopy with energy dispersive X-ray spectroscopy; Zn = Zinc. Elements that were detected above the limits of quantification are indicated in bold fonts for the total reflection X-ray fluorescence (TXRF).

The measurements showed that, for the CFM56-5B aircraft, the PNC and to a lesser extent the $d_{mn50}$ varied as a function of the engine rpm. Four different engine rpms were evaluated, including taxiing, idling flight, the maximum flight, and take-off speeds. During aircraft taxiing, the PNC was ~$10^7$ p/cm$^3$ with a $d_{mn50}$ at 16 nm. At idling flight speed, the PNC increased to $1.4 \times 10^7$ p/cm$^3$ with a $d_{mn50}$ at 17 nm, and again to $2.2 \times 10^7$ p/cm$^3$ with a $d_{mn50}$ at 15 nm at the maximum flight speed. Finally, the highest PNCs were found during take-off and estimated at $2.3 \times 10^7$ p/cm$^3$ with a $d_{mn50}$ at 15 nm. As for the GE90 reactor, the SEM-EDX analyses for the CFM56-5B aircraft engine demonstrated that the nanometric fraction was mainly composed of carbon and oxygen as a result of fuel combustion, but sodium and sulfur were also detected. As for the GE90, titanium was identified by TXRF at both the micrometric and submicronic scales in the CFM56-5B engine exhaust. This element could therefore serve, though not at the nanometric scale, as a tracer for reactor exposure. Zinc was also identified for CFM56-5B by TXRF as it was present at the submicron scale, in line with previous reports [19].

Figure 2 shows the PNC and $d_{mn50}$ for pushback tractors. The PNC values of $5.0 \times 10^5$ p/cm$^3$ and $2.4 \times 10^6$ p/cm$^3$ were recorded at idle and high speeds, respectively, for the new generation Pushback tractor 270. In contrast, the old generation Pushback tractor 256 had a similar PNC means between the idle and high speeds, with concentrations that could reach $3.0 \times 10^6$ p/cm$^3$. Nevertheless, a higher value spread was measured with the old generation. Such generational differences in emissions were also observed for $d_{mn50}$. The latter, recorded for the new generation tractor, varied in the range of 20–37 nm at idle speed and 28 nm at high speed. Concerning the old generation tractor, the $d_{mn50}$ recorded was extremely noisy with variations in the range of 20–101 nm at idle speed and in the range of 18–83 nm at high speed. Overall, the results show that the exhaust $d_{mn50}$ was higher for an old versus new generation tractor. The $d_{mn50}$ recorded for pushback tractor outlets exceeded those recorded during aircraft engine tests. The elemental analyses indicated that the nanometric fraction was mainly the result of fuel combustion, with the usual identification of carbon and oxygen. Interestingly, titanium was identified at the micrometric scale by TXRF analysis only on the old generation tractor. Concerning

the SEM-EDX analyses, chromium was observed at the micrometric scale for the new generation tractor.

When testing the generator set, the maximum PNC recorded was $1.6 \times 10^6$ p/cm$^3$ when idling and $1.4 \times 10^7$ p/cm$^3$ at high power. The corresponding variations in $d_{mn50}$ were 69 nm and 59 nm.

The PNC for the light vehicle exhausts differed according to the type of fuel used (Figure 2). The older diesel van had a maximum PNC estimated at $10^7$ p/cm$^3$, while the more recent gasoline car had a lower maximum PNC between $10^4$ p/cm$^3$ and $10^5$ p/cm$^3$ (consistent with a previous study on gasoline engines [20]). In both cases, the engine speed did not seem to significantly impact concentration. Interestingly, the speeds tested for the vehicles were idling and the equivalent of 50 km/h (2000 rpm), which is the maximum speed allowed for this vehicle on the apron. Differences in emissions between vehicle types were also observed for $d_{mn50}$. For the diesel van, the $d_{mn50}$ was in the range of 22–42 nm. The $d_{mn50}$ recorded for the gasoline car was in the range of 12–24 nm, which is lower than the diameters reported in previous studies [20,21]. This is probably due to differences in the motor operating point and type. The elemental analyses, integrating tests on both vehicle types, showed that the nanometric fraction was mainly the result of fuel combustion with the majority presence of carbon and oxygen. The SEM-EDX analyses indicated the presence of chromium at the micrometric scale.

### 3.2. Aerosol Measurements at the Medium-Haul Area Workstation

Measurements were taken at the medium-haul area during the "hub" period, a high activity period with incoming travelers gathered for afternoon long-haul flights. The period included the plane's arrival (11:07 a.m.), passenger and luggage disembarkation, maintenance tasks on and around the aircraft before new passengers and luggage board, and ultimately the departure of the aircraft (12:05 p.m.). Figure 3 focuses on the PNC and $d_{mn50}$ of aerosols during the parking of an A321 Airbus aircraft at the F30 parking.

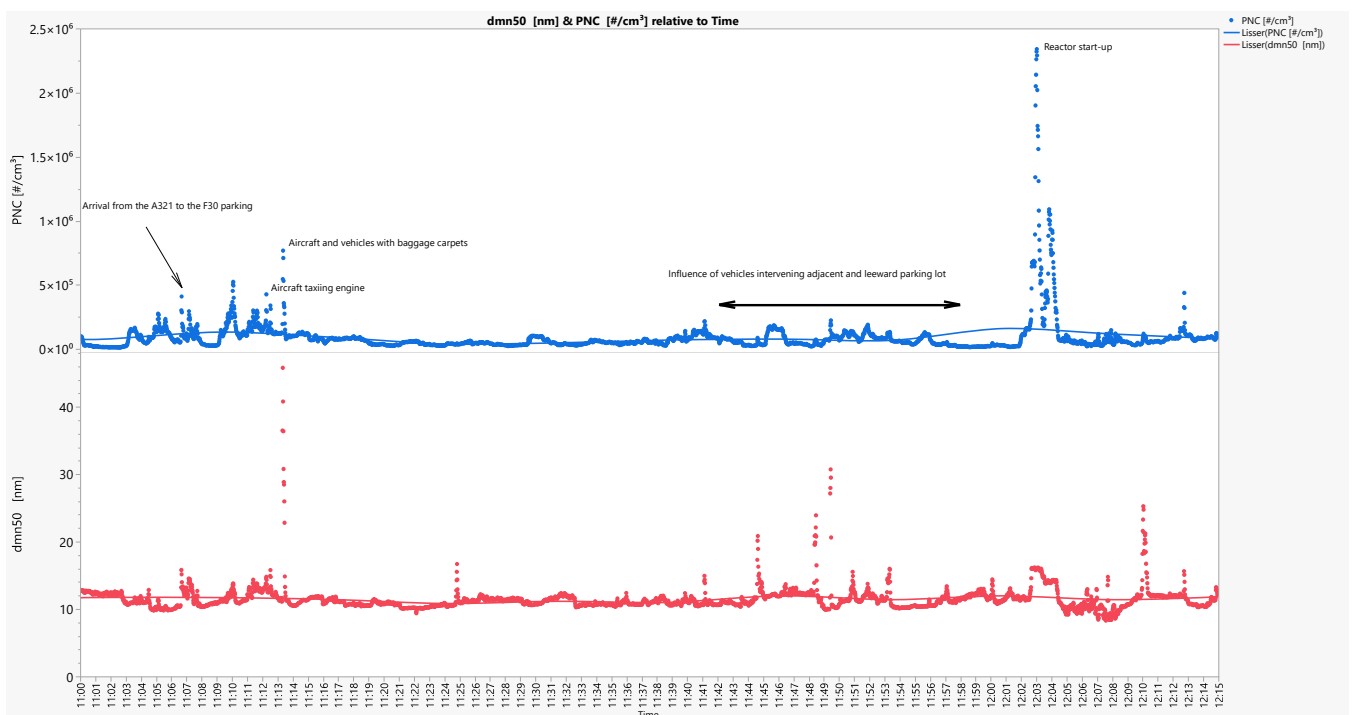

**Figure 3.** Variation in the concentration (PNC; blue curve) and median particle size ($d_{mn50}$; red curve) recorded by the fast mobility particle sizer spectrometer (FMPS) during the A321 aircraft parking at F30 of terminal 2F at the Paris-CDG airport.

The average PNC recorded on the apron was $1.2 \times 10^5$ p/cm$^3$ with a $d_{mn50}$ of 12 nm. When an emission $> 5.0 \times 10^5$ p/cm$^3$ was observed, it was due to the operation of an aircraft engine present in the near field or to the passage of an aircraft within ~100 m. The highest emission observed occurred during jet-engine start-up, with a concentration increase of $2.3 \times 10^6$ p/cm$^3$, i.e., a factor of 19 compared to the average apron PNC. This is consistent with findings at other airports [2,5,21,22]. Other activities taking place during aircraft stopovers (e.g., luggage unloading) are difficult to identify if the only information available is the PNC.

The minimum and maximum values recorded by the aethelometer for EC concentrations on the apron were, respectively, 0.9 μg/m$^3$ and 3.40 μg/m$^3$ with the average at 1.8 μg/m$^3$. The average ambient EC concentration (medical building: 4.8 μg/m$^3$) was higher than the average concentration recorded at the apron level (Table 2). The carbon value at the medical building was obtained with the Particlever® sampler (ITGA Group, Saint Gregoire, France) using the integrative filter sampling method presented in another paper on the evolution of the respiratory function of workers [14].

**Table 2.** Elemental carbon concentration (EC) and chemical tracer elements at the medium-haul aircraft area. Elements that were detected below and above the limits of quantification are indicated in standard and bold fonts, respectively, for the total reflection X-ray fluorescence (TXRF).

| Source of Emission | Aethelometer at the Medium-Haul Area | Integrative Filter Sampling at Ambient Concentration (Distance 5 km) | TXRF Chemical Identification at the Medium-Haul Area | | SEM-EDX Chemical Identification at the Medium-Haul Area | |
|---|---|---|---|---|---|---|
| | [EC] (μg/m$^3$) | [EC] (μg/m$^3$) | Micrometric Size | Submicronic Size | Micrometric Particles | Nanometric Particles |
| Medium-haul aircraft area | 1.8 ± 0.6 | 4.8 ± 0.6 | **Ca**, **Fe**, K, S, Si, Ti | Br, **Ca**, Zn | Al, Ca, Mg, O, Si | Ca, C, O, S, Si |

Al = Aluminum; EC = elemental carbon; Br = Bromine; Ca = Calcium; Fe = Iron; K = Potassium; Mg = Magnesium; O = Oxygen; S = Sulphur; SEM-EDX = Scanning electron microscopy with energy dispersive X-ray spectroscopy; Si = Silicon; Ti = Titanium; TXRF = Total reflection X-ray fluorescence; Zn = Zinc. Elements that were detected above the limits of quantification are indicated in bold fonts for the total reflection X-ray fluorescence (TXRF).

The elemental analyses of the apron filter showed that the nanometric fraction consisted primarily of carbon and oxygen (resulting from combustion). In terms of metallic elements, titanium and zinc were identified.

## 4. Discussion

This study's strength lies in its approach, which allows for the independent analysis of various emission sources and a comprehensive evaluation of the aerosols present at an airport.

### 4.1. Source Emissions: Engine Exhausts

By conducting a focused analysis of various sources emissions (i.e., Aircraft engine GE90 engine, Aircraft engine CFM56-5B engine, Pushback tractor 256, Pushback tractor 270, generator set, gasoline car, and diesel van), their characteristics can be determined in relation to the PNC, $d_{mn50}$, and chemical composition, while considering the functional parameters of the engines, such as the rpm and generation.

Firstly, the results obtained as part of this study are in line with the results of a study carried out by the CEA lab in 2012 at the Paris-CDG and Marseille airports on similar motor exhausts (Figure S4). This shows that there has been no major change in the physical nature of the aerosols (PNC and $d_{mn50}$) emitted by the main emission sources of aerosols on the apron. Secondly, the analysis of $d_{mn50}$ shows a correlation between the size and the contribution of the amorphous carbon form. The $d_{mn50}$ values of the CFM56-5B engine, Pushback Tractor 256, Pushback Tractor 270, generator set, and diesel van decreased at

higher rpms (Figure 2) and may be related to the amorphous form of carbon at a low power, which evolves toward a "graphitic" form at higher rpms [23]. The PNC and $d_{mn50}$ for the diesel van are consistent with the experiments performed by Agarwal et al. [24]. Concerning other vehicles on the apron (pushback tractors and light vehicles), the impacts of vehicle generation or the fuel type used are significant for both PNC and $d_{mn50}$. Curiously, Corporan et al. [25] showed a contradictory increase in the particle diameter with the rpm and Rogers et al. [26] showed that such correlations are, in fact, related to the engine type. How $d_{mn50}$ changes with the engine rpm, therefore, requires further investigation [25].

The elementary analyses showed that the chemical composition was linked to the particle size and also offered some clues to make a link between metallic tracer elements and the sources of emission. Thus, the elementary analyses (Table 1) showed that the nanometric fraction of the aircraft emissions was mainly due to combustion with a majority presence of carbon/oxygen. The metallic elements that can be used as emission tracers occurred mainly at the micrometric scale. The elements identified as potential tracers of aircraft emissions were titanium, zinc, and possibly bromine. As noted by Masiol et al., zinc and titanium in aircraft engine exhausts come from engine component abrasion or trace metal impurities in the fuel [27]. Bromine was found to be emitted from aircraft engines and Pushback tractor 270, but no explanation was found in the literature as to the origin of this emission. Further investigations will be needed to identify bromine sources.

*4.2. Aerosol Measurements at the Medium-Haul Workstation Area*

As mentioned and demonstrated above, the apron at airports represents a significant and specific source of air pollutants, especially aerosols [1–4]. The data collected at the medium-haul area demonstrated that, when a PNC measured in the near field or on the apron was $>5.10^5$ p/cm$^3$, this emission could be related to an aircraft engine (Figure 3). The highest emission observed during the start-up of an aircraft engine on the apron was ~19× higher than the average apron concentration. In addition, a number of activities taking place during an aircraft stopover can be identified based on the measurement of the median particle size ($d_{mn50}$). Cross-examining the $d_{mn50}$ variations alongside changes in the PNC helps to discriminate if an emission is due to a vehicle or an aircraft. For example, in Figure 3, at 11:13 am, the recorded emission can be attributed to a vehicle carrying luggage and not to an aircraft engine with an increase in the $d_{mn50}$ of above 20 nm (also see the particle size distributions, Figure S5). Moreover, lower concentration peaks are observed from 11:43 am to 11:55 am and are attributable to vehicle activity in the adjacent aircraft parking thanks to the $d_{mn50}$, which shows that it exceeds 20 nm at the time of the peaks. The median particle size ($d_{mn50}$) helps to confirm whether an emission peak is due to a reactor ($d_{mn50}$~15 nm) or to the activity of a vehicle ($d_{mn50} > 20$ nm). These results are in accordance with those of Austin et al. [28], who demonstrated a higher emission rate of small 10–20 nm sized particles for a landing aircraft. Thus, the $d_{mn50}$ can be used to confirm whether an emission peak is due to a reactor ($d_{mn50}$ ~15 nm) or to vehicle activity ($d_{mn50} > 20$ nm). The airborne $d_{mn50}$ is ~12 nm, which is characteristic of volatile organic compound nucleation in the environment [24] and is smaller than other classical areas of UFP emissions, like freeways [3,28,29]. Indeed, Lobo et al. [30] demonstrated that secondary particles were the main contributor to airborne number-based emissions. The measurements performed at medium-haul areas indicated that the aerosol particle size distributions (PSDs) in terms of numbers was highly specific for apron level exposure, which was mostly <100 nm. This specificity in the $d_{mn50}$ and the transitory aspect of the aircraft emissions in number on the medium-haul zone led to the results observed on the mass concentration in carbon. Thus, the EC concentrations obtained are of the same order of magnitude between the apron (1.8 μg/m$^3$) and 5 km away at the medical building location (4.8 μg/m$^3$) (Figure S3). This finding is consistent with the observations of Riley et al. [31], indicating that the EC concentrations are elevated near airports up to a distance of 10 km. Furthermore, the concentration level recorded at the apron of EC (1.8 μg/m$^3$) is consistent with the levels reported by other studies that focused on EC. Shirmohammadi

et al. [2] reported EC concentrations ranging from 1.4 µg/m$^3$ to 3.84 µg/m$^3$ at several American airports.

In terms of micrometric chemical elements, titanium and zinc identified at the micrometric scale could be used as tracers of airport activity on the apron (as also found during the engine tests and in agreement with results observed at other airports [27,32]). The nanometric fraction deposited on the Sioutas$^®$ membranes was mainly due to combustion with a majority presence of carbon/oxygen. This differentiation can also be observed microscopically, with a different morphology between combustion aerosols consisting of aggregates/agglomerates of nanometric particles and micrometric-sized particles (Figure S6).

### *4.3. Exposure Assessment of Aerosols*

Emissions at the airport are very specific, with PSDs remaining mostly <250 nm, $d_{mn50}$ < 100 nm, airborne $d_{mn50}$ at the medium-haul apron ~12 nm, and a high PNC emission value and $d_{mn50}$ ~15 nm for aircraft engines. This information is particularly valuable for the occupational assessment of workers working in aircraft emission zones. Indeed, this study shows that workers on airport aprons are exposed to high levels of UFPs produced by engines. However, the high concentration of UFPs has a low impact on the mass exposure measurements. As a result, the mass-based measurements carried out in a parallel study on workers did not enable us to discriminate between different groups of workers (directly exposed to emissions on the apron and not directly exposed) [14]. However, exposure to combustion UFPs has been identified as a health issue [6]. Therefore, it appears appropriate to carry out a study to monitor workers on the apron based on PNC and $d_{mn50}$ measurements. The transitory character of aircraft emissions (a few minutes) and the fact that, in an 8 h working day, the effective time on the apron for a worker directly exposed is approximately 4 h [14] would probably require workers to be instrumented for longer than a working day. The studies conducted by Iavicoli et al. [33], Asbach et al. [34], and Jagatha et al. [35] will aid in the design of such a research project.

### 5. Conclusions

This study offered a better comprehension of airport global atmosphere and the different particulates and UFP emission sources that compose it. Indeed, based on the standard operating procedures for characterizing aerosol emissions from 5 nm to 8 µm, a fine mapping of the different aerosol sources was conducted and allowed us to list the characteristics of each source in terms of the PNC, $d_{mn50}$, and metallic composition. This study highlighted that these characteristics may be affected by intrinsic engine parameters, such as speed or (new vs. old) generation. On the one hand, aircraft engine emissions lead to the highest emissions (linked to the engine speed) of UFPs, up to $10^7$ p/cm$^3$, and a $d_{mn50}$ of less than 20 nm. On the other hand, the engines of the various vehicles used on the apron have their own specific emissions (linked to the generation and fuel used), but overall, these emissions are lower ($10^4$–$10^6$ p/cm$^3$), with a higher $d_{mn50}$ from 20 to 100 nm.

To conclude, an interpolation between this mapping and the overall aerosols measured in an airport apron allows the defining of the main contributors' sources. As observed during the medium-haul measurement, the average PNC is around $10^5$ p/cm$^3$ and, on the basis of a cross-reading of the PNC and the change in the $d_{mn50}$, one-off emissions can be attributed to each of the possible contributors. The highest recorded emission ($2.3 \times 10^6$ p/cm$^3$), linked to a fluctuation in $d_{mn50}$, is a perfect illustration of this phenomenon. It can be compared to the start-up of an aircraft engine.

**Supplementary Materials:** The following supporting information can be downloaded at: https://www.mdpi.com/article/10.3390/air2010005/s1. Figure S1: Aircraft engine sampling location with: (a) the engine test area, (b) the schematic view of the Zephyr Air-France test bench with the location of the aerosol sampling point in red, (c) the aerosol sampling probe inside the exhaust chamber, and finally (d) the trolley mounted with the aerosol devices. Figure S2: Measurement location at the medium-haul area. Figure S3: Positioning of the ambient measurements (red circle) and measurements at the medium-haul area (blue circle) at the Paris-Charles de Gaulle airport. Figure S4: Box plots of particle number concentration (PNC) and the median particle size (dmn50) for the motor exhausts studied in 2012 at the Paris-CDG and Marseille airports. Figure S5: Discrimination of the type of emissions based on FMPS PSDs. Figure S6: SEM pictures of the atmospheric particles collected by Sioutas® on the apron at the medium-haul area: (a) micrometric particles observed at a magnification of 9 k, and (a) aggregate of nanoparticles observed at a magnification of 45 k. Table S1. Basic meteorological data, temperature (°C), pressure (hPA), humidity (%), rain (mm in 1 h), wind speed (10 m meters high), and wind direction (on 360°)—recorded at the Roissy meteorological station, obtained from https://meteo.data.gouv.fr (accessed on 19 February 2024, Latitude: 49°00′55″ North, Longitude: 2°32′04″ East), during the characterization of the exhaust engines (GE90, CFM56-5B, Pushback tractor 256, generator set, gasoline car, and diesel van) and at the medium-haul aircraft area.

**Author Contributions:** S.A. conducted, investigated, and supervised the aerosol measurements at the airport and wrote the original draft. D.L. and S.J. prepared and conducted with S.A. the research at the airport. C.P., E.Z., A.T. and C.S. revised and structured the paper. C.S. proposed Figure 1. E.Z. worked on the methodology design and provided additional data from a previous airport study in the Supplementary Materials to improve the discussion. L.T. supervised the Nanero study, which combined aerosol measurements and individual monitoring at the airport. S.C., expert on aerosol measurements, reviewed the paper and provided advice to improve it. All authors have read and agreed to the published version of the manuscript.

**Funding:** This research was supported by the French National Research Program for Environmental and Occupational Health of Anses (PNREST Anses, EST/2017/1/185). The study funder was not involved in the study design; in the collection, analyses, and interpretation of data; in the writing of the report; and in the decision to submit the article for publication.

**Data Availability Statement:** Data is contained within the article or Supplementary Materials Access to raw data of devices is restricted to consortium members.

**Acknowledgments:** The authors would like to thank all the Air-France staff members (administrators, nurses, etc.) who contributed to the success of this study and the Air-France workers who agreed to carry the sampling equipment.

**Conflicts of Interest:** Carey M. Suehs reports a previous grant and fees from AstraZeneca and a grant from GSK outside of the present work. The remaining authors have no competing interests to declare.

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
