# Peer review of "Emission Characteristics and Potential Exposure Assessment of Aerosols and Ultrafine Particles at Two French Airports"

_2813-4168, doi:10.3390/air2010005_

Round 1

Reviewer 1 Report

Comments and Suggestions for Authors

Author Response

Dear

Sir/Madam, Please find attached our reply.

Best regards.

Reviewer 2 Report

Comments and Suggestions for Authors

Minor editing of language is suggested to improve the quality of the manuscript.

Which two French airports were investigated for these observations? In the aims, only CDG Paris airport is mentioned. PNC median particle size were measured at which activity? time of the day? and season? Figure 1 is comprehensive but it must be on the same page for better explanation. detailed source sampling and aerosol sampling be provided Does engine exhaust varies from different aircraft? and how were these compared? Which metals were detected in the elemental emissions ? Table 2; How did they find the values? and how did they compare? Discussion needs further improvement by digesting the earlier cited references.

Comments on the Quality of English Language

Minor editing of language is suggested to improve the quality of the manuscript.

Author Response

(The authors gave the same response as above.)

Reviewer 3 Report

Comments and Suggestions for Authors

Why is the possible determination of bromine in the abstract?

Justify the choice of Titanium, Zinc and possibly Bromine as tracers of this type of pollution.

Table 2: 1.8 ± 0.6 (not ;)

Author Response

(The authors gave the same response as above.)

Reviewer 4 Report

Comments and Suggestions for Authors

This study examined the Particle Number Concentration (PNC), the median particle size, and the metallic composition of medium-haul area and engine aerosols at two French airports. Also this study quantified and showed chemical elements (Titanium, Zinc and possibly Bromine) were identified as potential tracers of aircraft emissions and occurred mainly at the micron scale. However, some critical information is missing, and several issues need to be revised. I recommend the manuscript be revised before being accepted for publication.

1. In lines 42-52 of the Introduction, it is too concise and would be better demonstrated by adding a summary of some relevant literature, then the motivation of the present study could be better preformed.

2. In line 73, "particle number concentrations" is represented by "PNC", and in lines 102-103, "particle number concentrations" is represented by "PNCs". Also, "particle size distributions" is represented by "dmn50" in line 73, and "the median particle size" is represented by "dmn50" in line 15. Please give a uniform expression for the abbreviation of the term.

3. In lines 146-147, "the Aethalometer was out of range during the measurement". Does this problem affect the results of the paper. It is necessary for the authors to give a more detailed explanation.

4. Some problems with the use of statement tenses: "4nm to 3µm in size are determined" in line 103 and "recordings were processed using" in line 108. It is essential to maintain consistency in formatting throughout the paper.

5. In the aerosol particle size distribution measurements, is a drying device added to the inlet of the instrument?

6. In Table 2, there is an error in the symbol of ";" in "1;8 ± 0;6", please change. Paying attention to the rigors of essay formatting cannot be ignored either.

7. What are the the limits of quantification of the chemical tracer elements used in the manuscript, please add.

8. In line 227, "substantially enhanced" is too vague, and it would have been clearer if a comparison in terms of data had been given. This makes the paper more rigorous and academic.

Comments on the Quality of English Language

Minor editing of English language required

Author Response

(The authors gave the same response as above.)

Reviewer 5 Report

Comments and Suggestions for Authors

Dear Authors,

The reviewed manuscript entitled “Emission characteristics and potential exposure assessment of  aerosols and ultrafine particles at two French airports” is very interesting.

In my opinion, the article may be allowed to be published after minor revision.

To improve quality of the paper I propose to take into attention following issues:

1)     Very important when studying the number and concentration of particles in the air considering the source of emissions is the distance from the source. It would be beneficial to supplement the information on the distances that the authors adopted, taking into account the emission sources studied (aircraft engines, pushback tractors, cars). This is, in addition to meteorological conditions, crucial bearing in mind the results obtained, as well as the possibility of replicating such studies by other research teams at other airports. This allows the results of such studies to be compared. Without this information, this is limited.

2)     It would also be beneficial to supplement information on meteorological conditions on the day of measurements. In particular, the results of measurements carried out in the air are affected by wind speed, which has an impact on the dilution of the pollutants studied. What was the background concentration of pollutants in the ambient air?

3)     It would be beneficial for readers to provide a summary with conclusions based on the analyses.

Minor comments:

Table 2. Table 2. For value Aethelometer at medium-haul, replace semicolons with a period.

Line 140: remove the space before the parenthesis "Figure 2 )"

Line 199: remove the comma in the caption "generator, set"

 Best regards

Author Response

(The authors gave the same response as above.)
